# Maternal Immune Activation and Enriched Environments Impact B2 SINE Expression in Stress Sensitive Brain Regions of Rodent Offspring

**DOI:** 10.3390/genes14040858

**Published:** 2023-04-01

**Authors:** Troy A. Richter, Ariel A. Aiken, Madeline J. Puracchio, Ismael Maganga-Bakita, Richard G. Hunter

**Affiliations:** 1Department of Psychology, Developmental and Brain Sciences Program, University of Massachusetts Boston, Boston, MA 02125, USA; 2School of Arts & Sciences, Massachusetts College of Pharmacy and Health Sciences, Boston, MA 02125, USA

**Keywords:** transposon, retrotransposon, epigenetics, early life stress

## Abstract

Early life stress (ELS) can have wide-spread neurodevelopmental effects with support accumulating for the idea that genomic mechanisms may induce lasting physiological and behavioral changes following stress exposure. Previous work found that a sub-family of transposable elements, SINEs, are repressed epigenetically after acute stress. This gives support to the concept that the mammalian genome may be regulating retrotransposon RNA expression allowing for adaptation in response to environmental challenges, such as maternal immune activation (MIA). Transposon (TE) RNAs are now thought to work at the epigenetic level and to have an adaptive response to environmental stressors. Abnormal expression of TEs has been linked to neuropsychiatric disorders like schizophrenia, which is also linked to maternal immune activation. Environmental enrichment (EE), a clinically utilized intervention, is understood to protect the brain, enhance cognitive performance, and attenuate responses to stress. This study examines the effects of MIA on offspring B2 SINE expression and further, the impact that EE, experienced throughout gestation and early life, may have in conjunction with MIA during development. Utilizing RT-PCR to quantify the expression of B2 SINE RNA in the juvenile brain of MIA exposed rat offspring, we found dysregulation of B2 SINE expression associated with MIA in the prefrontal cortex. For offspring experiencing EE, the prefrontal cortex exhibited an attenuation of the MIA response observed in standard housed animals. Here, the adaptive nature of B2 is observed and thought to be aiding in the animal’s adaptation to stress. The present changes indicate a wide-spread stress-response system adaptation that impacts not only changes at the genomic level but potentially observable behavioral impacts throughout the lifespan, with possible translational relevance to psychotic disorders.

## 1. Introduction

Long dismissed as genomic “junk”, transposable elements (TEs) are currently enjoying a period of renewed interest [1,2,3]. TEs comprise approximately half of the mammalian genome. Their discoverer, Barbara McClintock, noted in the 1950s that these elements were crucial contributors to an organism’s ability to deal with environmental stress [4,5]. This notion has gained support in the realm of stress neurobiology, with transposons having notable adaptive and potentially deleterious effects. Findings have associated their dysregulation in humans with post-traumatic stress disorder and in animal models of stress disorders, as well as a number of neurodegenerative and neuropsychiatric disorders [6,7,8,9,10,11,12,13,14,15].

TEs have been observed to have a number of genomic and epigenetic functions. A prime example is the TE subfamily of short-interspersed nucleic elements (SINEs), which have a high GC content making them essentially ‘hotspots’ for DNA methylation which is used by cells to suppress transcription and silence expression of nearby genes [16,17]. In rodents, B2 SINEs have the ability to prevent heterochromatin silencing of the developmental expression of genes while their primate-specific orthologue, Alu, also function as transcriptional enhancers or promoters [18,19,20]. Functionally, it is important to remember that the impacts of SINEs on transcription occur not only through DNA elements but also through their non-coding RNAs (ncRNA). Though they are typically silenced in somatic tissues, in response to stressors such as heat shock, SINE Pol II promoters are activated, and SINE RNAs are upregulated. This stress-mediated regulation occurs in both primate Alu and rodent B2 RNAs and acts to inhibit the transcription of most genes, except those that would be up-regulated during heat shock, through binding to Pol II [21,22].

Globally and selectively, applied acute stress was found to increase methylation levels at TE loci as well as to downregulate TE RNA expression [23]. Further work found that acute restraint stress was found to have repressive epigenetic effects in the hippocampus, notably through H3K9 and H3K27 methylation associated with the repression of retrotransposable elements, also coinciding with a reduction of coding and non-coding RNA that is typically repressed by DNA [24,25]. Recently, however, B2 elements were released from these repressive marks after treatment with high levels of corticosterone suggesting stress’s regulatory effects appear to be mediated via the glucocorticoid receptor, which interacts in a number of ways with the epigenome [26,27,28]. Through repeated stress, this repression decreases, suggesting that the transposable elements may impair genomic stability under conditions of chronic stress.

In humans, early life stress exposure can increase vulnerability to long-term behavioral abnormalities [29,30,31,32]. Stress experienced in utero is implicated in the development of several disorders in humans that may not be evident until prepubertal or adult stages [33,34,35,36], leaving its full impact on offspring unknown for a considerable portion of their lifespan. Impairments can include HPA axis disruption [37,38,39] and increased vulnerability to developing psychiatric disorders, including depression and schizophrenia [40,41,42,43]. This highlights a need to understand the connection between dysregulation in the stress response and development of psychiatric disorders later in life.

A well-documented method of inducing ELS in animal models is maternal immune activation (MIA), the process of introducing an immune challenge during the gestational period. We have previously shown that MIA alters the expression of a critical component to proper adaptation to stress, *FKBP5*, in stress sensitive regions of the rodent brain, especially the mPFC [44]. MIA can be involved in social and cognitive impairments of offspring in later life [45,46,47], including impaired object recognition [48,49], spatial memory [45,50,51], and social behaviors [45,52,53,54]. Notably, neuronal changes reported following MIA present similarly to the pathogenesis of schizophrenia and autism, including diminished NMDA receptor function, disrupted dopamine regulation, reductions in glycoprotein reelin, and general immune system impairment [45,55,56]. Beyond this, the larger interest is behavioral or biological changes in the offspring of mothers that experience MIA, how these experiences impact their development, and if disruptions can be mediated or reversed.

Early life environments have been found to impact adaptive skills in animals [57,58] following stress and other adverse events. Implementation of enriched environment (EE) paradigms can mediate or reduce some negative adaptations to stress in both early life and adulthood. Resiliency against ELS can be mediated by experiences such as EE, even dampening schizophrenia symptoms that typically present in late adolescence [59]. Enrichment can improve learning and memory performance, [60,61] and enhance social play behavior [62,63]. For lab animals, the EE paradigm typically includes a larger home cage with novel toys that vary in size, shape, and texture to increase an animal’s opportunity for exploration, activity, and social interactions facilitated through colony nesting [45,64,65,66,67], thus ultimately promoting brain development, plasticity, and cognitive performance [68,69,70,71,72]. Positive effects of EE observed at the epigenetic level provide additional insights into how interaction with one’s environment supports and promotes brain function throughout the lifespan [73].

With the overwhelming majority of EE research being implemented alongside animal models of maternal separation, prenatal restraint stress, early drug exposure, and the stated risks of MIA as a method of ELS, the need for research understanding potential stress response mediators following MIA arises. MIA’s pathogenic link to neurodevelopmental disorders can be explored through genome–environment interaction.

We started with the simple question of whether MIA affects the expression of B2 SINE specifically across stress-sensitive regions in the brain. Further, thanks to the implementation of the EE paradigm, we expanded our questioning to examine if enrichment will change the expression response of our TE of interest. Overall, this work sets out to identify the influence of MIA and EE on the expression of B2 SINE offspring in our rodent model, working with the following hypotheses: (1) MIA will induce changes in B2 expression in selected stress-sensitive brain regions and (2) EE will attenuate target expression responses in the stress-sensitive brain regions following MIA.

## 2. Materials and Methods

### 2.1. Animal Handling

Animal rearing, MIA, and EE protocols were carried out and recorded at MCPHS University following the procedures detailed by [45], and outlined briefly here. Male and female Sprague–Dawley rats were obtained from Charles River and housed at 20 °C on a 12 h light/dark cycle with food and water available ab libitum. Female rats were pair-housed in either animal care control (ACC) standard housing, social control (SC) grouped housing (animals housed in groups of two litters), or in environmental enrichment (EE). The EE housing consisted of a multilevel cage, twice-weekly toy changes, and ramps, with two dams raising their litters together. ACC dams raised their litter alone. On gestational day 11, dams were treated with 100 ug/kg of lipopolysaccharide (LPS: Escherichia coli, serotype 026:B6; L-3755, Sigma, St. Louis, MO, USA; 100 µg/kg) or a pyrogen-free saline i.p. injection. On postnatal day 22 (P22), one male and one female from each litter were euthanized, their brains removed, and immediately frozen on dry ice.

### 2.2. Brain Region Extraction

Frozen whole brains were received at University of Massachusetts Boston, USA and stored at −80 °C until use. The total number of male and female offspring brains received for analysis is detailed in Table 1. Brains were mounted and cut on a cryostat referencing landmarks in the and Khazipov and colleagues atlas (2015) [74]. A Harris Uni-Core 2.00 mm brain punch tool was used to extract the following regions at approximately every 200 µm section interval: prefrontal cortex (PFC), dorsal striatum (DST), nucleus accumbens (NAcc), medial pre-optic area (MPOA), dorsal hippocampus (DHipp), and ventral hippocampus (VHipp). Tissue from all six regions was extracted from the brain of each animal and stored in separate 1.5 mL Eppendorf tubes at −80 °C until it was processed for analysis.

### 2.3. RNA Isolation

Zymo Research *Quick*-RNA miniprep kits (Irvine, CA, USA) were used following standard manufacturer instructions to obtain isolated RNA. RNA measures were determined by NanoDrop 2000 (ThermoFisher, Waltham, MA, USA) and recorded. RNA samples were stored at −80 °C.

### 2.4. Reverse Transcription

Applied Biosystems High-Capacity cDNA Reverse Transcription kit (Waltham, MA, USA) was used following standard manufacturer instructions for cDNA synthesis. DNA measures were determined by NanoDrop 2000 (ThermoFisher, Waltham, MA, USA) and recorded. Each cDNA sample to be used for PCR was normalized to a concentration of 1 µg/µL and stored at 4 °C. Remaining cDNA was stored at −20 °C.

### 2.5. Polymerase Chain Reaction

An established lab protocol was followed for qPCR [75]. Reaction volumes are as follows: 5.5 µL cDNA at 1 µg/µL OR 5.5 µL nuclease free water (control wells), 7.5 µL Thermo Fisher PowerUp Sybr Green (Waltham, MA, USA), 1 µL forward primer, and 1 µL reverse primer. B2 SINE primers were purchased from Integrated DNA Technologies (San Diego, CA, USA) (FWD: AGATGGCTCAGCGGTTAAGA, REV: GACACACCAGAAGAGGGTATCA). Each reaction was run in duplicate alongside a GAPDH control on a 96-well plate. Applied Biosystems StepOnePlus System (Waltham, MA, USA) used for PCR and standard manufacturer run parameters were set.

### 2.6. Statistics

Statistics were performed using the software package Statistical Software for the Social Sciences (SPSS) version 27.0 (IBM, Armonk, NY, USA). While all analyses included both male and female animals, sex was not evaluated as the dataset was not adequately powered to examine sex differences. The 2^∆∆Ct^ method was followed to calculate the fold change of target expression in the samples. Target expression data were analyzed using two-way ANOVAs with the factors ‘gestational treatment’ (LPS vs. Saline) and ‘housing’ (ACC vs. EE). Probabilities were set at a level of *p* = 0.05. Post hoc tests were conducted using pairwise t-tests and Levene’s test (applied in the occurrence of unequal variances). All data are graphically expressed as mean ± SEM.

## 3. Results

### 3.1. Analysis for Each Brain Region of Interest

The medial prefrontal cortex was studied due to it being central to the behavioral and physiological responses associated with stress [76,77,78]. Within the mPFC, no interaction of treatment by housing was observed (F (2,74) = 0.531, *p* = 0.590). A main effect of gestational treatment was observed (F (1,74) = 4.316, *p* = 0.041) and a main effect of housing was also observed (F (2,74) = 3.854, *p* = 0.026). Tukey’s post-hoc test revealed that ACC rats differed significantly from EE rats (*p* = 0.044) while there was no statistically significant difference in expression between ACC and SC or SC and EE rats (*p* > 0.05) (Figure 1).

The dorsal striatum (DST) was examined as it is also highly stress-sensitive [79]. In the DST, there was no treatment by housing interaction effect (F (2,75) = 0.760, *p* = 0.472), no effect of treatment (F (1,75) = 0.734, *p* = 0.394), and no effect of housing (F (2,75) = 0.687, *p* = 0.506) (Figure 2).

The nucleus accumbens was selected due to involvement in stress-related alterations along with being implicated in observed schizophrenic disturbances [80]. In NAcc, there was no interaction (F (2, 75) = 0.889, *p* = 0.416), no effect of treatment (F (1, 75) = 0.400, *p* = 0.529), and no effect of housing (F (2, 75) = 0.889, *p* = 0.416) (Figure 3).

The medial preoptic area, which has a role in physical responses and the relaying of stress response pathway [81], was also chosen to be examined. In MPOA, there was no interaction effect (F (2, 75) = 0.405, *p* = 0.668), no effect of treatment (F (1,75) = 0.305, *p* = 0.582), and no effect of housing (F (2,75) = 0.405, *p* = 0.668) (Figure 4).

Finally, the hippocampus was selected for analysis for its broad reaching impacts on behavior, memory, and anxious responses [82,83,84,85,86,87,88,89]. In dorsal hippocampus, there is no interaction effect (F (2,74) = 0.026, *p* = 0.974), no effect of housing (F (2,74) = 0.026, *p* = 0.974), and no effect of treatment (F (1, 74) = 2.379, *p* = 0.128). In the ventral hippocampus, there was no interaction effect (F (2,75) = 1.843, *p* = 0.166), no effect of treatment (F (1,75) = 0.001, *p* = 0.982), and no effect of housing (F (2,75) = 0.894, *p* = 0.414) (Figure 5 and Figure 6).

### 3.2. Summary of Gestational LPS Impacts Offspring B2 Expression

Data analysis revealed that the LPS treatment administered at gestational day 11 does significantly alter B2 expression in the medial prefrontal cortex of offspring at P22 compared to control despite housing conditions.

### 3.3. Summary of Housing Environment on B2 Expression of Offspring

In comparison to the standard housing group (ACC), offspring that were reared in the enriched housing group (EE) did have a significant change in expression of B2 SINE at P22 in the medial prefrontal cortex, suggesting that an enriched environment can protect against expression of B2 SINE.

## 4. Discussion

With the intent of uncovering whether MIA supplemented with EE will impact the expression of the B2 SINE RNA in the brain, our work here revealed fascinating new observations while also highlighting some attractive avenues for future discovery. First, in addressing the questions of does MIA via LPS treatment impact regulation of B2 SINE and to exactly what extent it may contribute, the data demonstrate that LPS treatment administered during the prenatal period does affect B2 expression in the medial prefrontal cortex of offspring at P22. In the EE offspring’s mPFC, there was a significant effect of gestational treatment and housing condition. In terms of MIA, dams that received an injection of LPS show a greater expression of B2 SINE RNA compared to the dams that receive saline. This suggests that regardless of housing condition, an immune insult, like that of MIA, will upregulate the expression of the stress responsive ncRNA B2 SINE in the offspring. Bartlett et al. observed something similar in the hippocampi of rats that were injected with corticosterone and in cells treated with corticosterone [28]. Our study builds upon that study by investigating the persisting effects of MIA on the expression of this stress sensitive TE in pups, something that has not been shown before in this context. Additionally, we previously have shown *FKBP5* is significantly upregulated after MIA in the mPFC in the same animals, suggesting that MIA is affecting the expression of stress responsive genes in the same direction [44]. *FKBP5* is a co-chaperone of GR, which is a main player in the stress response, and B2 SINE RNA is a novel regulator of GR transcriptional activity, suggesting that this immune insult is changing the expression by somehow manipulating these stress responsive players over a long term [27]. In the same study mentioned previously, there were no significant differences in GR expression in the same brain regions examined here. This is likely due to the nature of timing in this study and GR expression recovered when examining the brain at P22. This further suggests that MIA is conveying B2 SINE expression differences on a longer time scale than the typical stress response.

In terms of housing conditions, we see a significant decrease in the expression of B2 SINE RNA in the medial prefrontal cortex of pups in the EE condition compared to the ACC condition. This suggests that environmental enrichment can provide some sort of protective effect in halting the expression of B2 SINE RNA. The consequences of this halt in expression are not fully understood, but this data begins to tease apart how environmental enrichment can influence the expression of retrotransposons, especially stress sensitive B2. The inclusion of a social control housing group was used to separate the effects of purely social enrichment versus an environment geared toward cognitive enrichment. Further work will be needed to parse the relative influences of various environmental factors on B2 expression.

The potential ability of EE to “protect” against any changes in expression of the targets across multiple brain regions is the final question we asked in this study. Again, early life stress has been connected with EE effects, but here we have observed EE’s impact on MIA when analyzing the expression of B2 SINE. In every region except the medial prefrontal cortex, we did not observe a difference in the expression of B2 SINE across gestational treatment groups or housing groups. This can be attributed to the nature of these stress responsive molecules, and any involvement of B2 may simply not be as susceptible to environmental enrichment when an immune challenge is presented at an early stage of development or when enrichment is present only until P22. It is possible that if dams were immune-challenged at a different time during gestation, there may have been less or more of an impact on the observed expression. Another consideration would be the P22 stage which is very early in development, and it is plausible that the effects of EE on our targets could have taken more time or would be seen later in the lifespan.

In the brain, actions of retrotransposons can be observed through every stage of life. Contributing to neural diversity and survival, development and brain disease, the scope of the RNA genome’s role is continuously developing [90]. Insertions of RTEs can alter protein-coding and regulatory regions of the genome, impacting gene expression and a variety of cellular outputs [91,92,93,94,95,96,97]. Researchers have observed its roles in development, differentiation, chromosome imprinting, and regulation of epigenetic machinery [98]. Through the research discussed here, information is added to understanding their role in adaptations to stress and environments during early life, namely that they do in fact change in response to these factors. An observed upregulation was seen following LPS treatment in the mPFC, showing a sensitivity of RTE expression in offspring at P22 to the MIA stressor. As Alu elements play a critical role in the formation of neural networks, epigenetic mechanisms, and the regulation of processes throughout the brain, their role in human cognition can not be understated [99]. With extensive knowledge of the role of the mPFC and Hippocampus specifically in cognition, learning, memory, and a variety of other crucial neuronal processes and behaviors, the continued study of their role, especially in the early life stress response, is needed. The data showing significant modifications of B2, the rodent Alu highlight this role during development further. The finding that B2 SINE decreases in the EE group mPFC highlights the ability of EE to truly “protect” from some stress-induced biological and molecular responses. There is the question of the role that the TEs are potentially playing in neuronal differentiation which may lead to alterations in overall brain function in later life [100]. An increase in TE expression can induce transcriptional silencing through multiple mechanisms [10]. Indeed, the transcriptional silencing by retrotransposons may serve as a defense mechanism to stress, namely histone methylation, which is increasingly being recognized as responsible for TE action along with DNA methylation [101]. Stress and environmental pressures or changes can increase mutation rates that can be either adaptive or simply a by-product of the experience. TEs clearly are a key contributor to this concept and may even provide the genetic diversification that arises in natural populations due to these circumstances [102].

The need to address the potential presence of inherent individual differences is necessary with this project and the current understanding and directions within the field. ‘Gene x environment’ interactions are being increasingly studied as the environment may contribute to over 50% of behavioral variation in non-human primates and about 50% in humans [103]. The effects of some EE paradigms as observed here and by others show that some parameters may be influenced more than others. However, some argue that biological background has minor relevance to the effects of EE implemented over longer time periods or with “greater” enrichment provided, and that simple EE as employed here will not greatly vary across parameters [104]. The effects of EE shown at the epigenetic level, however, present how it may provide a mechanism for the environment to support and promote brain function throughout the lifespan [73]. Cues that may be altered due to stress can affect brain development as well as gene expression throughout life in an attempt to meet the demands of an organisms’ environment. If these alterations are positive or negative is something that continues to be studied in stress responsiveness behavioral tests in later life [105]. Still, developmental diseases such as autism and schizophrenia likely result from a combination of genetic factors and early life stress.

The final aspect to address is considerations of sex differences. While the data gathered was not analyzed to draw conclusions regarding differences in the male and female responses to LPS and EE across brain regions, this paradigm does require further investigation under a specifically designed and expanded sex-difference study. There is support for EE results depending on both time in enrichment as well as the sex of animals. For decades, there have been works in determining the extent to which sex plays a role in stress and environmental impacts on individuals [106], with results looking even further into social v. physical enrichments and their impact dependency on sex. The work done with colony nesting females showed a greater reduction in depression-like responses while males increased in anxiety-like behavior [107]. Both gender and early life experiences and stress are likely working in the modulation of expression of our targets and, potentially, behavior later in life.

## 5. Conclusions

Developmental diseases such as autism, schizophrenia, and bipolar disorder are being increasingly linked with an etiology of polygenic and environmental risk factors [6,15,108,109,110,111,112]. While full understanding of this spans multiple fields of research, the work detailed here adds to some schools of thought surrounding the underlying causes and mechanisms at work in these disorders. Insight is gained into B2 SINE’s presence in response to an immune challenge, a technique used to model prenatal stressors. The need for future behavioral testing is certainly justified as we know that structural or genetic changes do not always indicate functional or observable behavioral changes within a subject and throughout the lifespan. Data presented answer the intended questions and also provide solid ground to build a deeper understanding of the role of retrotransposons in early life stress response and adaptation. Their expression throughout the lifespan and potential impacts on offspring behavior are areas that should continue to be explored.

## Figures and Tables

**Figure 1 genes-14-00858-f001:**
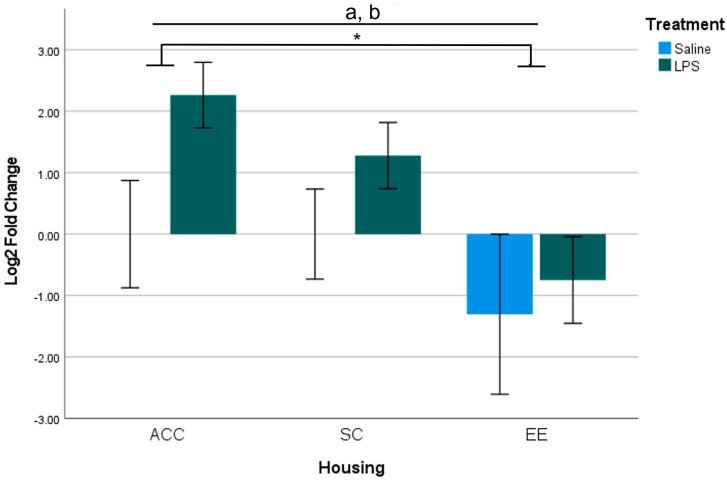
B2 SINE expression for Standard, Social Control, and Enriched housing conditions in the mPFC. Standard (Saline: n = 12, LPS: n = 10), Enriched (Saline: n = 12, LPS: n = 14), Social Control (Saline: n = 13, LPS: n = 15). A two-way ANOVA revealed simple main effects from housing and treatment. There were no statistically significant interactions. Post-hoc testing revealed a statistically significant difference between the enriched environment (EE) and the Standard Control (ACC). (* *p* ≤ 0.05, ^a^
*p* < 0.05, main effect of treatment, ^b^
*p* < 0.05, main effect of housing; mean ± SEM).

**Figure 2 genes-14-00858-f002:**
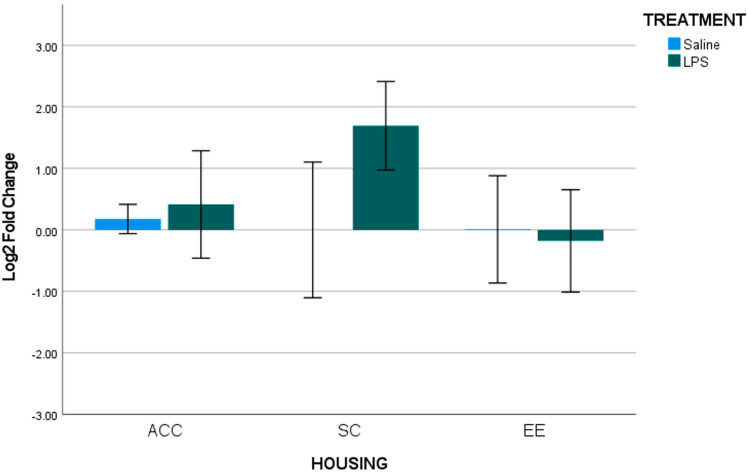
B2 SINE expression for Standard vs. Enriched and Standard vs. Social Control Housing in the DST. Standard (Saline: n = 12, LPS: n = 10), Enriched (Saline: n = 12, LPS: n = 14), Social Control (Saline: n = 13, LPS: n = 15). A two-way ANOVA revealed there were no simple main effects from housing or treatment. There were no statistically significant interactions.

**Figure 3 genes-14-00858-f003:**
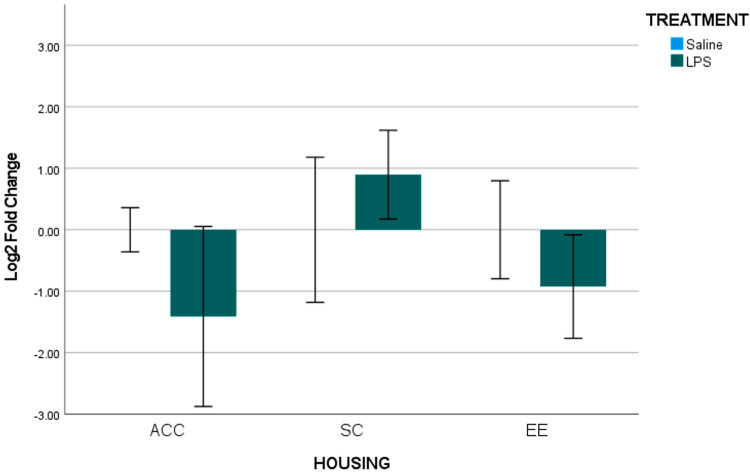
B2 SINE expression for Standard vs. Enriched and Standard vs. Social Control Housing in the NAcc. Standard (Saline: n = 12, LPS: n = 10), Enriched (Saline: n = 12, LPS: n = 14), Social Control (Saline: n = 13, LPS: n = 15). A two-way ANOVA revealed there were no simple main effects from housing or treatment. There were no statistically significant interactions.

**Figure 4 genes-14-00858-f004:**
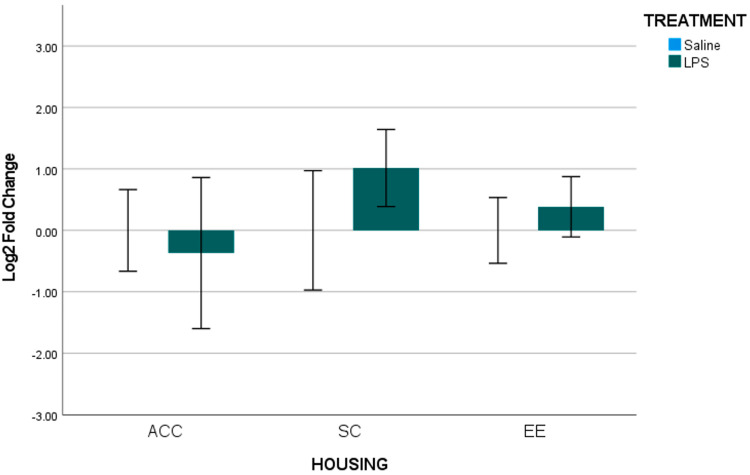
B2 SINE expression for Standard vs. Enriched and Standard vs. Social Control Housing in the MPOA. Standard (Saline: n = 12, LPS: n = 10), Enriched (Saline: n = 12, LPS: n = 14), Social Control (Saline: n = 13, LPS: n = 15). A two-way ANOVA revealed there were no simple main effects from housing or treatment. There were no statistically significant interactions.

**Figure 5 genes-14-00858-f005:**
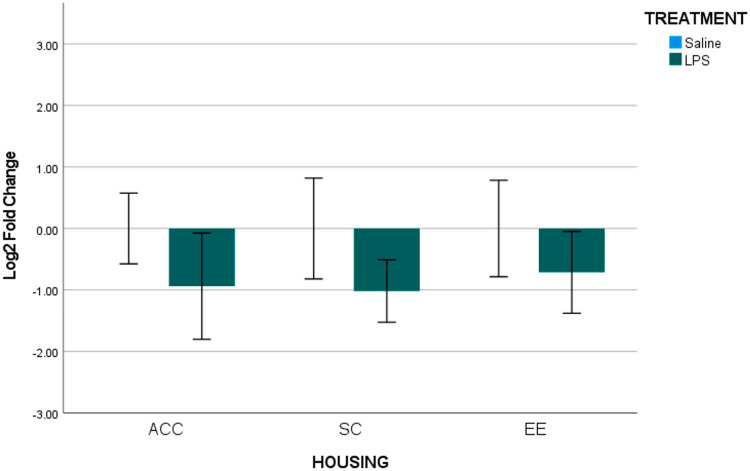
B2 SINE expression for Standard vs. Enriched and Standard vs. Social Control Housing in the Dorsal Hippocampus. Standard (Saline: n = 12, LPS: n = 10), Enriched (Saline: n = 12, LPS: n = 14), Social Control (Saline: n = 13, LPS: n = 14). A two-way ANOVA revealed there were no simple main effects from housing or treatment. There were no statistically significant interactions.

**Figure 6 genes-14-00858-f006:**
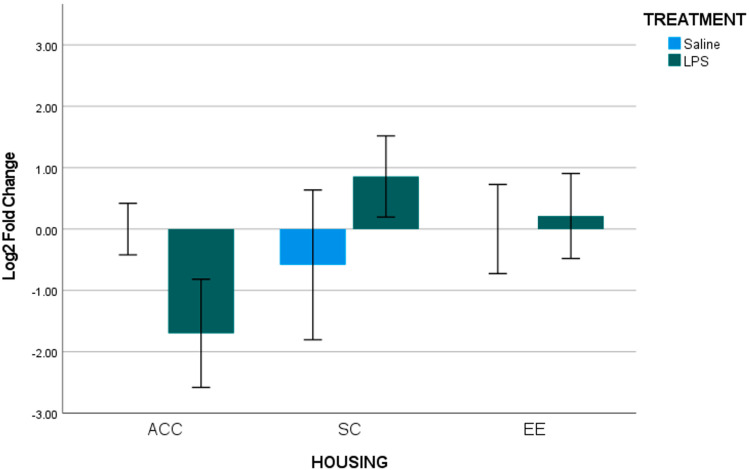
B2 SINE expression for Standard vs. Enriched and Standard vs. Social Control Housing in the Ventral Hippocampus. Standard (Saline: n = 12, LPS: n = 10), Enriched (Saline: n = 12, LPS: n = 14), Social Control (Saline: n = 13, LPS: n = 15). A two-way ANOVA revealed there were no simple main effects from housing or treatment. There were no statistically significant interactions.

**Table 1 genes-14-00858-t001:** Experimental Groups.

Housing Condition	N	Sex
Enriched Environment (EE)	28	13 Females and 15 Males
Social Control (SC)	27	12 Females and 15 Males
Animal Care Control (ACC)	23	13 Females and 10 Males

## Data Availability

The data presented in this study are available upon request from the corresponding author.

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
