# Peer review of "Maternal Immune Activation and Enriched Environments Impact B2 SINE Expression in Stress Sensitive Brain Regions of Rodent Offspring"

_genes, 2023, doi:10.3390/genes14040858_

Round 1

Reviewer 1 Report

In the manuscript by Richter et al, the authors sought to determine he impact of maternal immune activation (MIA) on B2 SINEs and the potential beneficial effects of environmental enrichment on B2 SINE expression. The goals of the current research are laudable given the recent invigoration of interest in transposable elements and their potential role as a genetic scaffolding for pathology development in response to environmental signals. To do this, the authors subjected rats reared in control, social, or enriched environments to MIA, dissected select brain regions implicated in pathology and measures B2 SINE RNA levels. The methods employed were sound. The choice of brain regions to measure seemed somewhat scattershot and did not appear to be focused on a particular form of pathology or circuit implicated in disturbed behavior. Nor was there any measure of behavioral consequences of MIA or EE to then related to changes (or lack of changes in B2 SINE levels). 

Despite those issues, the authors found that B2 SINEs are increased in PFC following MIA and that EE was associated with a reduction in B2 SINE (both relative to control and LPS challenged rats). Effects of the experimental manipulations on other brain regions were largely absent of effect. The authors interpret these results to indicate that TEs may be unregulated in response to challenge and that environmental manipulations (such as enrichment) could serve as a means to diminish such effects. Significant additional work will be required for bridging these effects with behavioral consequences, but the demonstration of regionally selective effects may be beneficial for guiding such studies. Thus, I don't have any significant concerns with the manuscript in its current form.  

Author Response

We thank this reviewer for his or her hard work reviewing our manuscript. We have made some minor edits to the manuscript.

Reviewer 2 Report

This is an interesting study that proposes MIA has the potential to influence stress pathways in offspring, especially B2 SINE activity. While the literature surrounding stress and abnormal expression of SINE is abundant, the conclusion in this study is not well supported by the data. It seems like RT-PCR of SINE B2 was the only molecular experiment that was conducted. A lot more could be done to make the claim more convincing, especially since the authors themselves mentioned stress and many epigenetic processes.

Major: 

  1. All data presented in the manuscript are results from RT-PCR of SINE B2; there is no data showing stress in the offspring. If stress is the cause for increased B2 SINE expression, it is crucial to show that the offspring presented signs of stress grossly or at the molecular level. Perhaps there is increased expression of CRH in the hypothalamus? Other evidence/data for cellular stress will compensate for this too.

  2. Only one brain region (mPFC) out of all the ones that were included showed a significant increase in B2 SINE expression. There is no discussion as to why mPFC is special in this sense. Again, maybe mPFC showed more stress response than other brain regions that didn’t have increased B2 SINE transcription?

  3. Graphs: it is better to present individual data points, especially because the SEM does not reflect data distribution at all. This decreases the statistical soundness of using ANOVA since it assumes that the data is normally distributed. This compounds the fact that only one brain region showed statistical difference

  4. There is no approved IACUC protocol number provided.

Minor: 

  1. Figure 1 would be better presented as a table.

  2. The introduction and discussion could be more succinct. In the intro, Hunter et al., 2012 paper showed lower TE expression—the opposite of what the manuscript is showing. Citing this article could be phrased differently (such as ‘controversial as to what the effect of stress is on TE expression…’) to make the intro appear more organized.

  3. The effect of sex was mentioned in the intro and discussion, but there is no mention of it in the results. The sex could be colored differently on the graph (or making dodged bar graph) and show that there is no difference.
